# Antimicrobial Efficacy of Silver Nanoparticles against Candida Albicans

**DOI:** 10.3390/ma15165666

**Published:** 2022-08-18

**Authors:** Razia Z. Adam, Saadika B. Khan

**Affiliations:** Department of Restorative Dentistry, Faculty of Dentistry, University of the Western Cape, Cape Town 7535, South Africa

**Keywords:** denture stomatitis, *Candida albicans*, dentures and denture liners, silver nanoparticles, colony forming units

## Abstract

Current treatment protocols for patients diagnosed with denture stomatitis are under scrutiny, and alternative options are being explored by researchers. The aim of this systematic review was to determine if silver nanoparticles inhibit the growth of *Candida albicans*, and the research question addressed was: *In adults, do silver nanoparticles inhibit the growth of Candida albicans in acrylic dentures and denture liners compared to normal treatment options*. A systematic review was the chosen methodology, and criteria were formulated to include all types of studies, including clinical and laboratory designs where the aim was tested. Of the 18 included studies, only one was a clinical trial, and 17 were in vitro research. The inhibition of candidal growth was based on the % concentration of AgNPs included within the denture acrylic and denture liner. As the % AgNPs increased, candida growth was reduced. This was reported as a reduction of candidal colony forming units in the studies. The quality of the included studies was mostly acceptable, as seen from the structured and validated assessments completed.

## 1. Introduction

Oral candidiasis is a fungal opportunistic infection [1]. The two most frequent oral diseases associated with Candida are denture stomatitis [2] and refractory root canal infection. Denture stomatitis (DS) is a form of oral candidiasis. It is reported that 70% of all complete denture wearers will suffer from denture stomatitis, with a female predilection [2]. The main commensal organism is *Candida albicans*, which is normally found in the oral cavity [3]. However, an imbalance may result in a proliferation of the organism, thereby increasing its pathogenicity. *Candida albicans* is a biphasic fungus that is present in two forms: yeast and hypha [4]. It is usually present on the mucus membrane in the yeast form, but when it invades the tissues, it converts to hyphal form. This hyphal form has a stronger pathogenicity. The *Candida albicans* organism’s key virulence attribute is the ability to adhere to surfaces. This means that removal from the oral cavity through saliva and swallowing is ineffective. When a patient wears a dental prosthesis, adherence of the fungus is to the denture surfaces due to its porous nature and hydrophobicity [4].

Management of the different types of denture stomatitis is problematic, as recurrent infections are common, and no gold standard for treatment exists [5]. Routinely, once DS is positively diagnosed, topical antifungals are prescribed but are often ineffective because of poor patient compliance, continuous removal by saliva and swallowing, and the infected denture base contacting the inflamed tissues. Systemic antifungals are not often used, and there is the threat of antimicrobial resistance developing.

The search for alternative treatment protocols has now expanded to the application of nanotechnology in dentistry. A few studies have reported on the use of nanoparticles in dental biomaterials [6,7,8]. Predominantly metal and metal oxide nanoparticles have been used in dental biomaterials, as they are antibacterial. Silver nanoparticles (AgNPs) are commonly explored as they have good antibacterial properties, are non-toxic to humans, and may be cheaper to produce with a lesser risk of antimicrobial resistance [8]. However, there is a paucity of clinical and other research specifically focusing on the inclusion of silver nanoparticles in acrylic resin of dentures and denture linings. With an increase in the ageing population globally, this treatment approach for DS using silver nanoparticles could be essential in improving patient outcomes and quality of life of denture wearers. This alternative form of treatment could become the gold standard for treating this commonly found denture-related condition.

The purpose of this study was to determine if silver nanoparticles inhibit the growth of *Candida albicans* when included in acrylic dentures and in different denture liners (temporary or permanent types).

## 2. Materials and Methods

### 2.1. Reporting Format

The PRISMA (Preferred Reporting Items for Systematic Review and Meta-Analysis) criteria were used for the preparation of this review [9]. A protocol was developed and registered with PROSPERO Registration No: CRD42019145542 and with the institutional ethics committee (Registration No: BM20/4/1). Since then, a detailed protocol for this SR has also been published [10].

### 2.2. Research Question

The research question, “In adults, do silver nanoparticles inhibit the growth of *Candida albicans* in acrylic dentures and denture liners compared to normal treatment options”, conforms to the PICO (Participants, Intervention, Comparator, Outcomes) structure:

Population (P): Adult patients with denture stomatitis; Intervention (I): Denture liners or denture acrylic with silver nanoparticles; Comparison (C): Other treatment options (disinfection agents, antiseptic mouthwashes, chlorhexidine, antifungal agents, topical antifungals, and systematic antifungals); Outcomes (O): Resolution of DS, reduction in colony forming units (CFUs), and prevention of biofilm formation.

### 2.3. Eligibility Criteria

Inclusion criteria: (i) primary clinical and laboratory studies (in vitro and in vivo human studies) investigating the incorporation of silver nanoparticles in denture acrylics or denture liners; (ii) published between 2000 and 2020 in English. Exclusion criteria: systematic reviews, case reports, expert opinions, and book chapters.

### 2.4. Search Strategy and Study Selection

An electronic search was conducted in the Ebscohost, Pubmed, Wiley, and Scopus databases for articles published during the period 2000–2020.

The following basic search strategy was used and adapted across the specific databases: (dentures OR denture liners OR complete dentures OR denture acrylic OR resilient liners OR tissue conditioners) AND (silver nanoparticles) AND (Candida albicans) AND (clinical trials, in vitro and in vivo studies, longitudinal studies OR observational OR randomized clinical trials) AND 2000–2020. The two authors (R.Z.A.; S.B.K.) identified and removed any duplications using the Mendeley Reference manager system. These articles were then exported into RAYYAN QCRI, a web tool for conducting systematic reviews. Once the titles of articles were independently screened, the abstracts of potential inclusion articles were read as indicated in the PRISMA flow chart. Any queries around eligibility were resolved by discussion between the two authors. Subsequently, the full text articles were read to confirm inclusion.

### 2.5. Data Collection

Both authors extracted the following data independently from the full text articles, which were finalized for inclusion: (i) authors, country of publication, year of publication; (ii) study methods, participants, intervention, type of AgNPs, concentration of AgNPs, type of microbial testing; (iii) outcomes and experimental results. The outcome measures in this systematic review were as follows: resolution of denture stomatitis and size of the inhibition zone, number of colony forming units (CFUs), and changes in microbial growth and biovolume of viable cells.

### 2.6. Quality Assessment and Risk of Bias

A risk of bias tool for clinical studies and one for laboratory studies was used.

#### 2.6.1. Risk of Bias Tool for Clinical Studies

A risk of bias assessment of the included clinical studies which evaluated the use of silver nanoparticles in denture acrylic and denture liners for this SR was completed across the following six domains: random sequence generation, random allocation concealment, blinding, incomplete outcomes data, selective reporting of bias, and other bias [11].

After evaluation, the studies were classified for these domains as having a low, high, or unclear risk of bias [11].

#### 2.6.2. Quality of Assessment of Laboratory Studies

Two validated tools were customized by the authors (R.Z.A.; S.B.K.) to evaluate the included studies for this SR based on the following criteria: (i) standardization of sampling procedures, (ii) description of sample size, (iii) calibration of sample before applying the test in accordance with standards and specifications such as International Standard Organization (ISO), (iv) evaluation of results, and (v) the use of appropriate statistical analyses [10].

### 2.7. Data Synthesis

A narrative and tabular synthesis was completed for all studies. Due to the quality, the heterogeneity, and the statistical analyses presented in each of the included studies, a meta-analysis could not be performed [11].

## 3. Results

The PRISMA flow chart as seen in Figure 1 represents the screening and selection stages of the studies to determine suitability for inclusion using a study eligibility form and to answer the research question for this systematic review [10].

### 3.1. Study Selection

Four hundred and fifty-eight articles were identified and retrieved through the stipulated electronic databases searched (Figure 1). Using Mendeley to de-duplicate records, the titles of four hundred and eleven articles remained to be screened. Of the records that were screened, three hundred and sixty articles were excluded, as these did not meet the inclusion criteria. Only twenty-six full text articles met the prerequisite criteria according to their abstracts. After detailed evaluation of the full text articles using the study eligibility form created for this SR, the final number of included studies was eighteen, as eight were excluded, and reasons for their exclusion will be reported on (Figure 1).

### 3.2. Study Methods and Characteristics

The final 18 included studies were all published between 2008 and 2019, highlighting the period of nanoparticle research, which was mostly reported in the last few years. The studies included were conducted in 11 countries only and mostly in third world and developing countries, with only a few from developed nations: Brazil [12,13,14,15]; China [16,17]; Egypt [18]; India [19]; Iran [20,21]; Italy [5]; Japan [22]; Korea [23,24,25]; Mexico [26]; Poland [27]; and Serbia [28], as seen in Table 1.

In terms of study design, only one clinical trial was conducted where there was inclusion of nanoparticles to the denture acrylic, and therefore it was included [18] in this SR. The rest of the studies (*n* = 17) for this SR were laboratory or in vitro studies (Table 1).

Figure 1 below clearly indicates the stages of identification, screening, and inclusion with the expectant searching results and elimination using stipulated criteria and the final number of included articles relevant for this SR.

Table 1 below illustrates the demographic and basic descriptive information related to all the included studies; further details are shared on the other tables and within the narratives.

Table 1 above only illustrates different types of acrylic resins (chemical and tradenames) where nanoparticles were included and tested. In addition, the different organisms, including candida, are also only mentioned on Table 1.

The addition of silver nanoparticles in both denture acrylic and denture liners, the subject of this SR, was investigated differently across the included studies [5,12,13,14,15,16,17,18,19,20,21,22,23,24,25,26,27,28] A variety of different denture acrylics used in dentistry are reported in this SR. More importantly, different formulations of the AgNPs, which were manufactured differently and included within these denture acrylics, were tested to see the impact on candidal growth, as seen in Table 2 below.

Acosta Torres et al. (2012) [26] was the only study that used plants to synthesize silver nanoparticles, while all the other studies either purchased silver nanoparticles [18,28] or synthesized silver nanoparticles chemically [5,12,13,14,15,16,19,20,21,22,25,27]. The methods used for the chemical synthesis of AgNPs also varied greatly amongst the different studies. The size of the silver nanoparticles was not reported in all studies [12,15,16,17,18,22,25,28], and where included, it ranged from 5 nm to 120 nm (Table 2).

As these were mostly in vitro studies, it cannot be reported if AgNPs may cause resolution of DS. However, from the one clinical study focusing on complete dentures, it was observed that after 4 weeks, all fungal species disappeared, where AgNPs at a 0.2% concentration were included in the acrylic [18]. The % concentration of AgNPs included ranged from 0.5% to 30% weight and indicating a decrease in CFUs as the % weight of AgNPs was increased, irrespective of the formulation and synthesis of the AgNPs.

Table 2 also indicates how an increase in % weight of AgNPs causes a reduction in CFUs in small or large sample studies after varied lengths of incubation periods. The above descriptions imply that irrespective of all the differences across studies related to manufacturing of AgNPs, size of the sample or AgNPs, and types of AgNPs or acrylics, a reduction in CFUs is seen that is only dependent on % weight of AgNPs increasing. AgNPs have been shown to impact the inhibition of the growth of *Candida albicans*. However, as stated before, whether this implies a reduction or resolution of DS cannot be deduced and must still be investigated further, especially clinically and after using it on patients.

### 3.3. Quality Assessment and Risk of Bias

The quality of the 18 included studies was assessed, and the results are summarized in Figure 2 and Table 3 included below. The quality of a study, be it clinical or in vitro, greatly impacts the implementation of the outcomes of the research. The translation of good quality laboratory studies implies that the material may be ready to be tested clinically on patients.

The reporting in this section is divided into clinical and laboratory evaluations of the included studies.

(a)Clinical study (Figure 2)

The only clinical study, a randomized clinical trial, was assessed using a risk of bias for clinical studies [11,18] (Figure 2). The six domains of the Risk of Bias tool as observed in Figure 2 indicated a recording of “high risk of bias” across one domain only, that of the blinding of the outcomes assessor and a recording of “unsure” of the blinding of both the participants and personnel who partook in the study. This does impact on the quality of parts of the research only. For all other domains, namely randomization (sequence generation and allocation concealment), incomplete outcomes data and selective reporting, and other types of biases, the assessment recorded a “low risk of bias”, as the required information was available in the article. The fact that for five categories, the study had a score of “low risk for bias” implies that the study was conducted well, and that most categories were reported showing good quality research. Thus, overall, the quality of the clinical trial seems acceptable.

(b)In vitro Laboratory studies (Table 3)

The 17 in vitro studies plus the laboratory aspect of the clinical study were assessed using an instrument which was adapted by the authors by combining the tools used in two other studies [10] (Table 3). The scoring system for this adapted tool is detailed in a previous publication [10], but an explanation of the final scores is given here.

Of the 18 studies, only 2 had a final score of between 8 and 10, having a “high risk of bias” and indicating that most criteria for the risk of bias assessment were not met, making them studies of poor quality [5,28]. However, four of the included studies (23%) were recorded as having a “low risk of bias”, indicating that the criteria for this tool were largely met, and these were high quality studies [17,21,24,26].

However, the majority of included studies (64%; *n* = 12) recorded an assessment of “moderate risk of bias”, indicating that certain aspects of the research needed improvement [12,13,14,15,16,18,19,20,22,23,25,27]. Upon closer evaluation, aspects that needed development and maybe even an upgrade included the reporting of sample size calculation and sample calibration (Table 3). In summary, the studies reported for this SR were mostly of an acceptable quality, as standardizing of sampling and reliable measuring of outcomes were conducted.

### 3.4. Effectiveness of AgNPs against Candida Albicans in Denture Acrylic

Antimicrobial activity was measured in a variety of manners: agar diffusion [12,19,28], colony forming units [5,13,15,17,20,21,22,23,24,25], minimum inhibitory concentrations (MICs) [14,21], Cell Proliferation Kit II (XTT assay) [14,15,16], crystal violet assay [13], and microbial viability assay based on luminescent ATP measurement (Bac Titer-Glo™, Promega, Fitchburg, WI, USA) [26]. The time period tested also differed across the studies with it ranging from 1 h to 72 h. These different types of measurements could also impact on the quality of the outcomes measured for this SR.

The sample sizes across studies ranged from 18 to 360 specimens, and many other aspects such as manufacturing of AgNPs, size of the AgNPs, and types of AgNPs or acrylics, as stated before, indicated heterogeneity of the included studies, and thus it was not possible to combine information across studies to produce a meta-analysis. It was also not possible to make a uniform conclusion from these studies in terms of the incorporation of AgNPs because the studies were heterogenous on so many aspects, as mentioned above, including the different research tests and microbial species and strains tested for.

### 3.5. Effectiveness of AgNPs against Candida Albicans in Denture Liners

Only four studies investigated the effect of incorporating AgNPs into denture liners. Agar diffusion [12] and calculating colony forming units (CFUs) [21,23,27] were methods used to determine antimicrobial activity. The results of these studies are also shown in f 2.

The concentration of AgNPs used differed between studies, but they generally agreed that as the % AgNP increased, the effectiveness increased due to decreases in CFUs. Kreve et al. reported a similar antifungal effect at 5% and 10% [12]. In the study by Nam (2011), AgNP tissue conditioner samples showed minimal fungicidal concentration at doses above 0.5%, and no CFU was detected at 2.0% concentration of AgNP and above [23]. There was no statistical difference between 24 h and the extended 72 h incubation time before testing (*p* > 0.05) for the antimicrobial effect (Table 2). Mousavi et al. (2019) found that complete inhibition of *Candida albicans* occurred at 24- and 48-h intervals at 10% and 20% concentrations, although the best effects were achieved with a 5% concentration of ZnO-Ag at 24- and 48-h intervals [21]. Chladek et al. (2011) found that the in vitro antifungal efficacies (AFEs) of the samples were 16.3% to 52.5%. However, no statistical significance was reported [27].

### 3.6. Excluded Studies

The reasons for exclusion varied, but these were based on the inclusion criteria set for this SR, and these are as follows:testing was not completed on denture acrylic or denture liners [29,30]the study did not measure antimicrobial properties [31,32]the study was not testing silver nanoparticles per se [33,34]the excluded study did not look at removable but rather fixed partial denture prosthesis [35]

## 4. Discussion

The objectives of this review, namely resolution of DS, reduction in CFUs, and prevention of biofilm formation, were largely met even though certain aspects of the included studies are questionable. We found 18 articles reporting the incorporation of AgNPs into denture acrylics or denture liners, and of these, 17 were laboratory studies, where most indicated the inhibition of the growth of *Candida albicans* based on the % concentration of AgNPs included. Even the one clinical study indicated a positive impact of inclusion of AgNPs within the acrylic, even though it was evaluated only after 4 weeks. This could be explained by the well-established effectiveness of silver nanoparticles as an antibacterial agent due to their ability to selectively destroy cellular membranes.

However, no definitive conclusions about comparative efficacy could be drawn because these studies were highly heterogenous with regards to several aspects of the individual studies, as stated before, which is why a meta-analysis could not be completed. Amongst all these studies, no two studies used the same sources and ages of microorganisms, testing of antimicrobial activity, AgNPs synthesis or how these were used, incubation times, and periods it was tested. Having said this, AgNPs inclusion as a DS treatment alternative can still be considered, gauging the positive outcomes with laboratory studies. This highlights the necessity for testing its efficacy clinically on a much larger scale rather than only accepting the positive outcome from one clinical trial reported here.

The current SR included mostly laboratory studies, and even though most of these reported a decrease in CFU formation with the % increase in concentration of AgNPs included within the acrylic or liner material, the nature of this type of treatment protocol requires clinical evidence. Conducting more rigorous clinical studies is thus advised and re-emphasized as the next step. Then also, it is prudent to include plant-based or formed AgNPs within the acrylic or liner as they may be more appropriate for use in these forms and will provide better clinical evidence, as reported in the Acosta Torres study [26]. Green synthesis has been recognized for its use in biological applications, as it is easy to use and apply and is cost-effective [8,36,37]. This could be compared to current DS management protocols and the variations to address the concerns with effectiveness of treatments.

Translational research must be viewed as a step-by-step approach and should be conducted in phases, especially when new material efficacy needs to be proven still. Moreover, if the material use is geared to clinical practice and to change old clinical management approaches, it is important to ensure sufficient and quality evidence is provided. For this SR, AgNPs are seen as a new and alternative management approaches for DS, a commonly seen denture-related condition. The current approaches have been questioned, and for various reasons, mentioned above, alternatives such as AgNPs are being considered. Using the approach with the evidence presented in this SR, practitioners may be guided to include AgNPs in their daily management of patients presenting with DS sooner than later.

Quality research, both laboratory and clinical studies, must be the goal when conducting research. Conducting poor laboratory research, observed when doing quality assessments as observed with a few of the included studies for this SR, cannot even be used as a guide for clinical research. The evidence included in this SR was largely from good laboratory research, as not many clinical studies using AgNPs in patients with DS were found. However, translation of quality laboratory research to clinical studies, with such a viable alternative treatment option as the inclusion of AgNPs in the acrylic, should be considered. Similarly, poor clinical research cannot guide clinical practice either. The rigor of the new intended research must also be considered and cannot be over-emphasized. Thus, conducting randomized clinical trials will provide the ultimate quality clinical evidence that is needed to address the treatment protocol for DS, which can serve as a guide for changing the current status quo.

These SR outcomes also highlight the need for the development of guidelines for testing dental biomaterials with nanoparticles. This will ensure that the same information about the nanoparticles is reported on, such as its type, size, shape, and distribution within the biomaterials.

The SR conducted related to the inclusion of AgNPs in denture acrylic is significant, as it highlights the need for quality evidence when alternatives to current DS management approaches are sought. Moreover, the conversation to change current clinical DS management protocols has been started, and alternatives have been identified. However, sufficient clinical evidence is lacking, and the outcome of this SR regarding the resolution of DS cannot be definitively concluded. Thus, it is advised that inclusion of AgNPs and the forms as advised above be tested further clinically.

## 5. Conclusions

The inhibition of the growth of *Candida albicans* based on the % concentration of AgNPs included within the denture acrylic and denture liner were observed. That is, a reduction in candidal growth or colony forming units in both laboratory and clinical studies were identified as the key outcomes, irrespective of the variations amongst the studies. However, resolution of DS using AgNPs in the denture acrylic cannot be accepted as a definitive conclusion.

## Figures and Tables

**Figure 1 materials-15-05666-f001:**
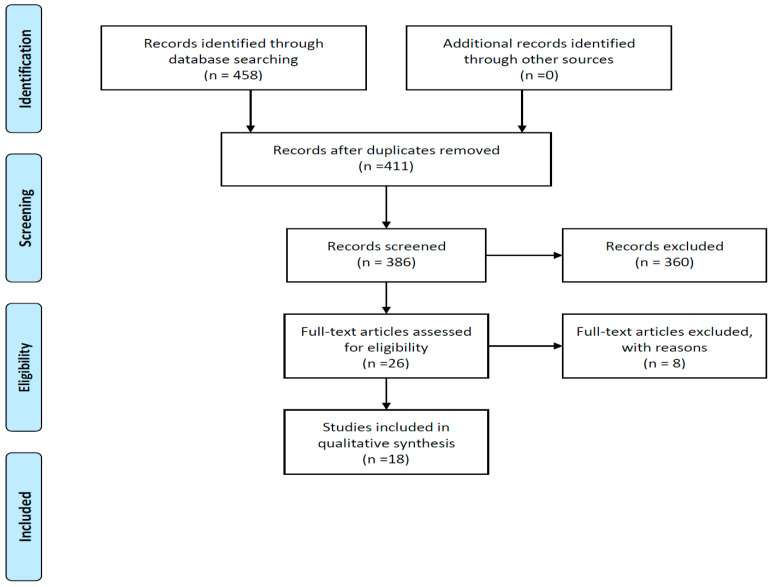
PRISMA flow diagram indicating record numbers for final inclusion (Moher et al., 2019 [9]).

**Figure 2 materials-15-05666-f002:**
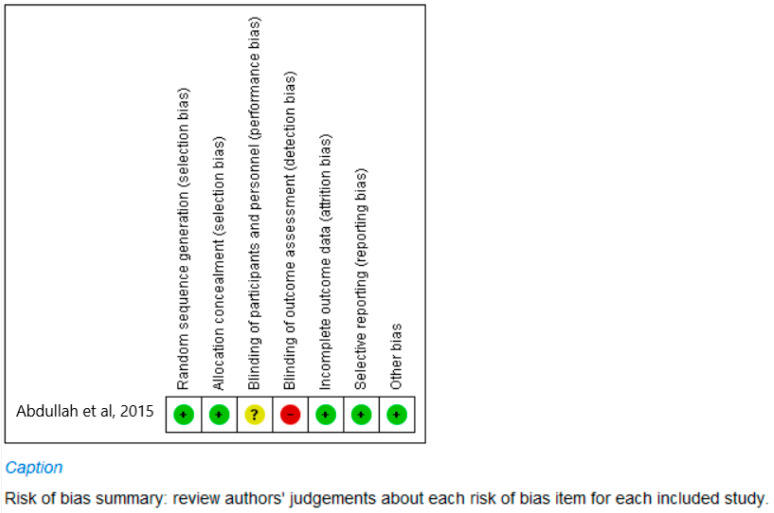
Risk of bias summary for included clinical study (Abdullah et al., 2015 [18]).

**Table 1 materials-15-05666-t001:** Descriptive details of included studies.

Study	Year	Country	Study Design	Type of Material	Organism
Abdallah et al., 2015 [18]	2015	Egypt	RCT	Heat-cured acrylic resin (Acrostone; Acrostone Dental factory, under exclusive license of England, Egypt)	Alternaria, Aspergillus, Aureobasidium, Candida, Cladosporium, Claviceps, Cryptococcus, Eurotium, Fusarium, Nigrospora, Pyrenophora, Saccharomyces, Schizophyllum, and Zygosaccharomyces.
De Matteis et al., 2019 [5]	2019	Italy	In vitro	Resin Paladon 65 (Kulzer)	*Candida albicans*
Kamikawa et al., 2014 [22]	2014	Japan	In vitro	Heat-cured acrylic resin for denture base, Acron	*Candida albicans* and *Candida glabrata*
Kreve et al., 2019 [12]	2019	Brazil	In vitro	Resin denture liner (Trusoft)	*Staphylococcus aureus* (ATCC 6538), *Pseudomonas aeruginosa* (ATCC 27853), *Candida albicans* (ATCC 90028), and *Enterococcus faecalis*
Lee et al., 2008 [25]	2008	Korea	In vitro	Heat-polymerized denture resin	*Candida albicans* (American type culture collection (ATCC) 66026)
Monteiro et al., 2014 [13]	2014	Brazil	In vitro	Denture acrylic resin	*C. albicans* (American Type Culture Collection (ATCC) 10231) and *C. glabrata* (ATCC90030), two Candida oral clinical isolates were used, namely, *C. albicans* 324LA/94 (obtained from the culture collection of Cardiff Dental School, Cardiff, UK) and *C. glabrata* D1
Wady et al., 2012 [14]	2012	Brazil	In vitro	Microwave denture base acrylic (Vipi wave)	*Candida albicans* strain ATCC 90028
Nam, 2011 [23]	2011	Korea	In vitro	GC Soft liner	*Candida albicans ATCC 14053*
Nam et al., 2012 [24]	2012	Korea	In vitro	Acrylic denture powder (Lucitone 199)	*Candida albicans* strain, ATCC 66026
Nikola et al. 2017 [28]	2017	Serbia	In vitro	Cold polymerized acrylic powder (Triplex Cold, Ivoclar Vivadent).	*Staphylococcus aureus* ATCC 25923 and fungus *Candida albicans* ATCC 2091
Acosta Torres et al., 2012 [26]	2012	Mexico	In vitro	Denture acrylic (Nature cryl)	*C. albicans* strain (90026)
Li et al., 2016 [16]	2016	China	In vitro	Denture base acrylic resin	*C. albicans* conference strain 3153A
Ghahremanloo et al., 2015 [20]	2015	Iran	In vitro	Denture acrylic resin	*Candida albicans* ATCC 10231; *Streptococcus mutans* ATCC 35668; *Candida albicans* hospital isolated
Suganya et al., 2014 [19]	2013	India	In vitro	Heat cure polymethylmethacrylate (DPI heat cure material)	*Candida albicans*
De Castro et al., 2016 [15]	2016	Brazil	In vitro	Dencor Lay Autopolymerizable (SC); and Clássico thermopolymerizable (TR) (Clássico Artigos Odontológicos1) acrylic resins were used	*C. albicans* (ATCC 10231) and *S. mutans* (ATCC 25175).
Chladek et al., 2011 [27]	2011	Poland	In vitro	(Ufi Gel SC): chemically cured silicone soft liner	*Candida albicans* (ATCC 10231)
Mousavi et al., 2019 [21]	2018	Iran	In vitro	Tissue conditioner GC soft liner	*S. aureus* (ATCC6538), *P. aeruginosa* (ATCC9027), *C. albicans* (ATCC10231), *E. faecalis* (ATCC29212)
Han et al., 2014 [17]	2014	China	In vitro	PMMA powder	*C. albicans* (76615), *S. mutans* (UA159)

**Table 2 materials-15-05666-t002:** Silver nanoparticle details as used in the included studies.

Study	Concentration of AgNPs	Size of AgNPs	Evaluation Method	Time Period	Sample Size
Abdallah et al., 2015 [18]	0.05 wt% and 0.2 wt%				30 patients
De Matteis et al., 2019 [5]	3 wt% and 3.5 wt%	20 nm	Miles and Misra assay (CFU)	24 and 48 h	Not indicated
Kamikawa et al., 2014 [22]	Not indicated		SEM and CFU	1, 3, 8, and 12 h	*n* = 40
Kreve et al., 2019 [12]	1, 2.5, 10%		Agar diffusion	24 h	*n* = 60
Lee et al., 2008 [25]	0.002, 0.01, and 0.05 M		CFU	24 h	No information
Monteiro et al., 2014 [13]	54 mg/L	5 nm	CV assay, total biomass, and CFU	5 h and 24 h	
Nam, 2011 [23]	0.1, 0.5, 1.0, 2.0 to 3.0 vol/vol%	100–120 nm	CFU	72 h	*n* = 162
Wady et al., 2012 [14]	1000, 750, 500, 250, and 30 ppm	9 nm	MIC, MFC, XTT	48 h (MFC, MIC); 3 h (XTT)	*n* = 360
Nam et al., 2012 [24]	1.0, 5.0, 10.0, 20.0–30.0 wt%		CFU	24 h	*n* = 90
Nikola et al., 2017 [28]	2, 5, 10%		Disc diffusion	24 h for bacteria; 48 h for fungi	
Acosta Torres et al., 2012 [26]		10–20 nm	Microbial cell viability assay based on luminescent ATP measurement (Bac Titer-Glo™, Promega, Fitchburg, WI, USA)	24 h	*n* = 18
Li et al., 2016 [16]				adhesion (24 h); biofilm assays (72 h)	
Ghahremanloo et al., 2015 [20]	2.5, 5, and 10 *w*/*w* %	22 nm	CFU	1, 6, 24 h	*n* = 160
Suganya et al., 2014 [19]	2.5%, 3%, and 5%	20–100 nm	Agar diffusion/CFU	24 h	*n* = 40
De Castro et al., 2016 [15]	0.5, 1, 2.5, 5, and 10% wt%		XTT and CFU	48 h	*n* = 80
Chladek et al., 2011 [27]	10, 20, 40, 80, 120, and 200 ppm	10–30 nm	CFU, AFE	17 h	
Mousavi et al., 2019 [21]	0.625, 1.25, 2.5, 5, 10, and 20 wt%	20 nm	MIC, CFU	24 and 48 h	*n* = 168
Han et al., 2014 [17]	1, 2, 3, 4, 5, 6 wt%		CFU	48 h	*n* = 36

**Table 3 materials-15-05666-t003:** Risk of bias for laboratory studies.

Scores for Risk of Bias Criteria							
		Standardization of Sampling Procedures	Description of Sample Size Calculation	Calibration of Samples before Testing\Standards	Measured Outcomes in Valid, Reliable Manner	Appropriate Statistical Analyses Used	Summary Score
1	Nam et al., 2012 [24]	0	2	1	0	0	3
2	Nikola et al., 2017 [28]	0	2	2	2	2	8
3	Acosta-Torres et al., 2012 [26]	0	0	0	0	0	0
4	Li et al., 2016 [16]	0	2	1	1	0	4
5	Ghahremanloo et al., 2016 [20]	0	2	2	2	0	6
6	Han et al., 2015 [17]	0	2	0	0	0	2
7	Suganya et al., 2014 [19]	0	2	1	2	2	7
8	de Castro et al., 2016 [15]	1	2	1	0	0	4
9	Chladek et al., 2011 [27]	0	2	2	1	2	7
10	Mousavi et al., 2019 [21]	0	0	1	1	0	2
11	Abdallah et al., 2015 [18]	0	2	2	1	0	5
12	De Matteis et al., 2019 [5]	2	2	2	2	2	10
13	Kamikawa et al., 2014 [22]	0	2	2	2	0	6
14	Kreve et al., 2019 [12]	1	2	1	1	0	5
15	Lee et al., 2008 [25]	2	2	0	1	0	5
16	Monteiro et al., 2014 [13]	0	2	2	2	0	6
17	Nam, K. Y. (2011) [23]	0	2	2	2	0	6
18	Wady et al., 2012 [14]	0	2	2	2	0	6

KEY: Low Risk of Bias: 0–3; Moderate Risk of Bias: 4–7; High Risk of Bias: 8–10.

## Data Availability

The files are available upon request.

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
