# Peer review of "Antimicrobial Efficacy of Silver Nanoparticles against Candida Albicans"

_materials, 2022, doi:10.3390/ma15165666_

Round 1

Reviewer 1 Report

 Antimicrobial efficacy of silver nanoparticles against Candida albicans

  1. The novelty of this is article has rarely stated. There are many papers are published in this filed, even, authors have already published a systematic review almost in the same topics. What is the difference between these two studies? Link: https://journals.plos.org/plosone/article?id=10.1371/journal.pone.0245811
  2. What was findings from your systematic review? It should be included in your abstract section. You have only mentioned the steps of your work in the abstract. Therefore, It is recommended that author should improve the abstract sections.
  3. The introduction section must be improved. The research gap of your study has not been identified clearly.
  4. There are several studies regarding the antimicrobial activity against the candida albicans. Why you select silver nanoparticles for this review? This must be included in your introduction sections.
  5. In page 5, there is a title of table, However no data. After that a figures. Then again a data. Please carefully check it and correct it. This is not understanding.
  6. There was no analysis regarding Table 1 as well as less descriptive for Table 2. You should analysis the findings in your texts.
  7. You must improve your writings. For example, in line 235, you used “But because the methodology” in the middle of the sentence. Please, carefully check the full manuscript and revise it accordingly.
  8. What is the effectiveness of AgNPs against Candida albicans in denture acrylic? You just wrote this the titles, however no analysis was mentioned regarding the effectiveness in your text.
  9. You are using same reference in one line by one line. Suppose in line 242-243, you use four reference and again in line 243-244 use the same reference. Multiple references are of no use for a reader and can substitute even a kind of plagiarism, as sometimes authors are using them without proper studies of all references used. In the case, each reference should be justified by it is used and at least short assessment provided. Please avoid lump reference in your full text of studies.
  10. In your conclusions, please discuss the implications of your revie. Discussions and conclusions must go deeper, it would be more interesting if the authors focus more on the significance of their findings regarding the importance of the interrelationship between the obtained results and the barriers to do it, what would be the consequences, in the real world, in changing the observed situation, what would be the ways, in the real world, to change/improve the observed situation.
  11. Reference style was not accurate. Follow the instructions.

Author Response

Please see the document attached.

Reviewer 2 Report

Dear authors,

after reading this manuscript, I suggest making some changes, in order to be more clear:

Why did you choose the time period: published between 2000-2020 in English

Why did you choose these databases: Ebscohost, Pubmed, Wiley and Scopus data- 80 bases for articles published during the period 2000-2020.

How did you interpret this statement: The size of the silver nanoparticles was not reported in all studies. Was it not important?

How do you comment the Risk of Bias summary for Included Clinical study

What relevance has your research if :” The sample sizes across studies ranged from 18 to 360 specimens but because the 235 methodology for the studies varied, it is not possible combine information across studies. 236 It was also not possible to make a uniform conclusion from these studies in terms of the 237 incorporation of AgNPs because the studies were heterogenous using different materials, 238 different research methods and different microbial species and strains.”?

And “However, no conclusions about comparative efficacy could be drawn because these stud- 279 ies were highly heterogenous, thus a meta-analysis could not be completed. Amongst all 280 these studies, no two studies used the same sources and ages of microorganisms, testing 281 of antimicrobial activity, AgNPs synthesis or how these were used, incubation times and 282 periods it was tested. Having said this, AgNPs inclusion as a DS treatment alternative can 283 still be considered gauging the positive outcomes with laboratory studies.” ?

How were the nanoparticles included in the acrylic?

Author Response

Please see document attached.

Reviewer 3 Report

I think the theme of the manuscript is good.

The content also supports the results.

However, I think some parts need retouching.

1. Please remove (1), (2), (3), and (4) from the abstract.

2. Please specify "A systematic review" in the abstract.

3. Please elaborate on "Results" in the abstract.

4. In Results:

3.2. Study methods and characteristics 

Please rewrite it because the expression is too unclear.

5. Discussion The antibacterial effect of silver nanoparticles can be added.

Author Response

Please see document attached.

Round 2

Reviewer 1 Report

manuscript significantly improved 

Reviewer 2 Report

Dear authors, congratulations on your work!

Hopefullty this research might have also have a clinical impact. 

Kind regards,